Correcting nuisance variation using Wasserstein distance

Tabak Gil tabakg@google.com
http://orcid.org/0000-0003-3422-0370 Fan Minjie
Yang Samuel
Hoyer Stephan
Davis Geoffrey
Google , Mountain View, CA , USA
Gomez Shawn
Electronic publication date: 2020 Feb 28
Publication date: 2020
Volume: 8
Electronic Location ID: e8594
Received 2019 Jul 10; Accepted 2020 Jan 18
Copyright: © 2020 Tabak et al.
Copyright year: 2020
Copyright holder: Tabak et al.
License: This is an open access article distributed under the terms of the Creative Commons Attribution License, which permits unrestricted use, distribution, reproduction and adaptation in any medium and for any purpose provided that it is properly attributed. For attribution, the original author(s), title, publication source (PeerJ) and either DOI or URL of the article must be cited.
License URL: https://creativecommons.org/licenses/by/4.0/

Keywords: Wasserstein distance, Cellular phenotyping, Batch effect, Embedding, Minimax, Optimal transport, Domain adaptation

Funding: Google LLC Funding was provided by Google LLC. The funders had no role in study design, data collection and analysis, decision to publish, or preparation of the manuscript.

==============================
Profiling cellular phenotypes from microscopic imaging can provide meaningful biological information resulting from various factors affecting the cells. One motivating application is drug development: morphological cell features can be captured from images, from which similarities between different drug compounds applied at different doses can be quantified. The general approach is to find a function mapping the images to an embedding space of manageable dimensionality whose geometry captures relevant features of the input images. An important known issue for such methods is separating relevant biological signal from nuisance variation. For example, the embedding vectors tend to be more correlated for cells that were cultured and imaged during the same week than for those from different weeks, despite having identical drug compounds applied in both cases. In this case, the particular batch in which a set of experiments were conducted constitutes the domain of the data; an ideal set of image embeddings should contain only the relevant biological information (e.g., drug effects). We develop a general framework for adjusting the image embeddings in order to “forget” domain-specific information while preserving relevant biological information. To achieve this, we minimize a loss function based on distances between marginal distributions (such as the Wasserstein distance) of embeddings across domains for each replicated treatment. For the dataset we present results with, the only replicated treatment happens to be the negative control treatment, for which we do not expect any treatment-induced cell morphology changes. We find that for our transformed embeddings (i) the underlying geometric structure is not only preserved but the embeddings also carry improved biological signal; and (ii) less domain-specific information is present.

Introduction

In the framework where our approach is applicable, there are inputs (e.g., images) and a map F sending the inputs to vectors in a low-dimensional space which summarizes information about the inputs. F could either be engineered using specific image features, or learned (e.g., using deep neural networks). We will call these vectors “embeddings” and the space to which they belong the “embedding space.” Each input may also have corresponding semantic labels and domains, and for inputs with each label and domain pair, F produces some distribution of embeddings. Semantically meaningful similarities between pairs of inputs can then be assessed by the distance between their corresponding embeddings, using some chosen distance metric. Ideally, the embedding distribution of a group of inputs depends only on their label, but often the domain can influence the embedding distribution as well. We wish to find an additional map to adjust the embeddings produced by F so that the distribution of adjusted embeddings for a given label is independent of the domain, while still preserving semantically meaningful distances between distributions of inputs with different labels.

The map F can be used for phenotypic profiling of cells. In this application, images of biological cells perturbed by one of several possible biological stimuli (e.g., various drug compounds at different doses, some of which may have unknown effects) are mapped to embeddings, which are used to reveal similarities among the applied perturbations.

There are a number of ways to extract embeddings from images of cells. One class of methods such as that used by Ljosa et al. (2013) depend on extracting specifically engineered features. In the recent work by Ando, McLean & Berndl (2017), a Deep Metric Network pre-trained on consumer photographic images (not microscopic images of cells) described in Wang et al. (2014) was used to generate embedding vectors from cellular images, and it was shown that these embeddings clustered drug compounds by their mechanisms of action (MOA) more effectively. See Fig. 1 for exemplary images of different MOAs.

Figure 1 Flowchart describing the procedure to generate image embeddings using a pre-trained Deep Metric Network and remove nuisance variation from them.

The embedding generator, denoted by F, is described in “Embeddings Based on Deep Neural Network”, which maps each 128 by 128 color image into a 192-dimensional embedding vector. The nuisance variation removal by our method is denoted by WDN (i.e., Wasserstein Distance Network). The 12 images on the right show representative images of cells treated with drug compounds with one of the 12 known mechanisms of action (MOA), from the BBBC021 dataset (Ljosa, Sokolnicki & Carpenter, 2012).

Experiments of this type are becoming increasingly common (e.g., see Caicedo et al. (2017)) as labs worldwide invest in high-throughput microscopy, and so we anticipate an increasing number of datasets of this type will become available. For this reason, there will be an increasing need for methods to obtain signal from this type of dataset.

Currently, one of the most important issues with using image embeddings to discriminate the effects of each treatment (i.e., a particular dose of a drug compound, the “label” in the general problem described above) on morphological cell features is nuisance factors related to slight uncontrollable variation in each biological experiment. Many cell imaging experiments are organized into a number of batches of experiments occurring over time, each containing a number of sample plates, which each contain individual wells in which thousands of cells are grown and treatments are applied. For this application, the “domain” is an instance of one of these hierarchical levels, and embeddings for cells with a given treatment tend to be closer to each other within the same domain than from different ones. As one example resulting in nuisance variation, the experimentalist may apply slightly different concentrations or amounts of a drug compound in two wells in which the same treatment was anticipated. Another example is the location of a particular well within a plate or the order of the plate within a batch, which may influence the rate of evaporation, and hence, the appearance of the cells therein. Finally, “batch” effects may result from differences in experimental conditions (temperature, humidity) from week to week; they are various instances of this hierarchical level that we will consider as “domains” in this work.

Many efforts have been made to correct for nuisance variation such as batch effects, especially for microarray gene expression data. The simplest method is data normalization such as mean-centering, standardization, and quantile normalization. However, data normalization is often not sufficient to ensure the correction of batch effects so more advanced methods have been developed. Alter, Brown & Botstein (2000) proposed a singular value decomposition based method that filters out the “eigengenes” (and “eigenarrays”) representing noise or experimental artifacts. Benito et al. (2004) used linear discrimination methods such as distance weighted discrimination to adjust for batch biases. Johnson, Li & Rabinovic (2007) proposed parametric and non-parametric empirical Bayes frameworks (i.e., ComBat) that remove the additive and multiplicative batch effects. Leek & Storey (2007) introduced Surrogate Variable Analysis to overcome the problems caused by heterogeneity in expression studies by identifying the effect of the hidden factors that may be the sources of data heterogeneity. Gagnon-Bartsch & Speed (2012) proposed the removal of unwanted variation method that restricts the factor analysis to negative control genes to infer the unwanted variation.

Most of the aforementioned methods are essentially coordinate-wise in the sense that the batch effect is assumed to affect each dimension (or more specifically each gene for gene expression data) independently. However, batch effects can be multivariate. This is especially true for the embeddings derived from cellular images. The embedding coordinates are often inter-correlated, and hence can be affected by batch effects jointly. Lee, Dobbin & Ahn (2014) proposed a multivariate batch adjustment method that can correct for the variance-covariance of data across batches. In particular, they derived an affine transformation that exactly matches the mean vectors and covariance matrices of two batches of data by assuming one of the batches as the target batch (or called the golden batch). The estimation of the transformation was based on a factor model and hard thresholding.

More recent work has started to remove batch effects using deep learning methods. Shaham et al. (2017) proposed a method that matches the distributions of data in source and target batches using a non-linear transformation based on residual networks, where the distance between the two distributions is measured by the MMD. This method has two distinct advantages compared to the previous work: it allows for non-linear removal of batch effects and matches the entire distribution instead of the first two moments for the two batches. Matching only the first two moments can be insufficient if the data distribution is multi-modal or highly non-Gaussian. We discuss this further in the appendix. For the embeddings of cellular images, this is likely to happen when individual cells are at different stages of the cell cycle or respond differently to a drug compound and hence form subpopulations. Another class of methods are based on the autoencoder. Amodio et al. (2018) learned a latent space of the data using autoencoder, identified batch effect related dimensions, and aligned the distributions of these dimensions across batches. Shaham (2018) used a variational autoencoder approach to learn a shared encoder used to obtain a batch-free encoding of the data, which contains solely biological signal, and also batch dependent decoders that allow for reconstruction of the data to ensure the entire true biological signal will not be lost or distorted. We further discuss these recent methods and how they compare to our method in “Conclusions.”

Caicedo et al. (2018) used a weakly supervised method to generate embeddings using the idea that the embeddings corresponding to similar treatments should be more similar. This is conceptually similar to the idea we use in our article, in the sense that certain embeddings should be closer to other embeddings. However, in this article the method is used during training, while we apply our method after a set of embeddings has already been found. Godinez, Hossain & Zhang (2018) used a similar approach with multi-scale convolutional neural networks, as well.

In this article, we address the issue of nuisance variation in image embeddings by transforming the embedding space in a domain-specific way in order to minimize the variation across domains for a given treatment. Our main goal is to introduce a general flexible framework to address this problem. In this framework, we use a metric function measuring the distances among pairs of probability distributions to construct an optimization problem whose solution yields the appropriate transformations for each domain. In our present implementation, the 1-Wasserstein distance is used as a demonstration of a specific choice of the metric that can yield substantial improvements. In particular, the 1-Wasserstein distance makes few assumptions about the probability distributions of the embedding vectors.

We highlight that other distances may be used in our framework, such as the Cramer distance or other Wasserstein distances. The Cramer distance may be preferable since it has unbiased sample gradients (Bellemare et al., 2017). This could reduce the number of steps required to adjust the Wasserstein distance approximation for each step of training the embedding transformation. Additionally, we discuss several possible variations and extensions of our method in “Conclusions.” However, the important properties of the Wasserstein metric that other alternatives should have are (1) is well-behaved for back-propagation and (2) takes into account higher-order moments. We discuss these reasons in more detail in “General Approach.”

Our method can incorporate all replicates of a given treatment on an equal footing, which can be advantageous over using only the controls when these replicates are available. Further, weights can be added if there is prior information about which distributions we want to match more strongly. For example, our process could be used on a per-treatment level (by modifying the weights of pairs selected for a given treatment) or a per-domain level (by modifying the weights of pairs having one of the two distributions belonging to a given domain).

Our source code for extracting the dataset and running the analysis is available at sourcecode.

Materials and Methods

Problem description

Denote the embedding vectors xt,d,p for t ∈ T, d ∈ D, and p ∈ It,d, where T and D are the treatment and domain labels respectively, and It,d is the set of indices for embeddings belonging to treatment t and domain d. Suppose that xt,d,p are sampled from a probability distribution νt,d. Our goal is to “forget” the nuisance variation in the embeddings, which we formalize in the following way. We wish to find maps Ad transforming the embedding vectors such that the transformed marginal distributions ν~t,d have the property that for each t ∈ T and di, dj ∈ D, ν~t,di≈ν~t,dj (for some suitable distance metric between distributions). Intuitively, the transformations Ad can be regarded as domain-specific corrections. This is based on the assumption that the embeddings of all the treatments in the same batch are affected by the nuisance variation in the same way. The transformations Ad should be small to avoid distorting the underlying geometry of the embedding space, since we do not expect nuisance variation to be very large. While optimizing the transformations Ad, we will at the same time be estimating pairwise distances between transformed embeddings from different domains with the same treatment (which we would like to minimize). This will lead to a minimax problem where we minimize over the transformation parameters and maximize over the distance-estimating parameters.

General approach

The 1-Wasserstein distance (hereafter will be simply referred to as the Wasserstein distance) between two probability distributions νr and νg on a compact metric space χ with metric δ is given by (1) W(νr,νg)=infγ∈Π(νr,νg)⁡E(x,y)∼γδ(x,y)

Here Π (νr, νg) is the set of all joint distributions γ (x, y) whose marginals are νr and νg. This can be intuitively interpreted as the minimal cost of a transportation plan between the probability masses of νr and νg. In our application, the metric space is Rn and δ is the Euclidean distance.

Ultimately our goal is to transform pairs of empirical distributions of embeddings so that they become indistinguishable. We use the Wasserstein distance towards that goal because of two reasons. The first is that when the Wasserstein distance between two distributions νr and νg is zero, they must be identical up to a set of measure zero. Because of this, two empirical distributions drawn from νr and νg cannot be distinguished by any classifier. The second reason is that during the optimization procedure, using the Wasserstein distance yields non-vanishing gradients, which are known to occur for metrics based on the KL-divergence, such as the cross entropy (Arjovsky, Chintala & Bottou, 2017). This is important from a practical point of view because vanishing gradients may halt the solving of the resulting minimax problem in our method. Specifically, we will be maximizing over a set of parameters to obtain estimates for the Wasserstein distance (instead of using a classifier) while at the same time minimizing over the transformation parameters. To highlight the vanishing gradient issue, consider the following case: If we had used a linear classifier in place of the Wasserstein distance, the classifier would converge very quickly, having vanishing gradients except near a hyperplane separating the initial empirical distributions, where the gradients become very large.

The usage of the Wasserstein distance can be extended to more than two distributions. Given two or more probability distributions, their mean can be defined based on the Wasserstein distance, known as the “Wasserstein barycenter.” Explicitly, the Wasserstein barycenter of N distributions ν1,…, νN is defined as the distribution μ that minimizes (2) 1N∑i=1NW(μ,νi)

The Wasserstein barycenter and its computation have been studied in many contexts, such as optimal transport theory (Cuturi & Doucet, 2014; Anderes, Borgwardt & Miller, 2016). In Tabak & Trigila (2018), the Wasserstein barycenter has been suggested as a method to remove nuisance variation in high-throughput biological experiments. Two key ingredients of the Wasserstein barycenter are that (i) the nuisance variation is removed in the sense that a number of distinct distributions are transformed into a common distribution, and hence become indistinguishable; and (ii) the distributions are minimally perturbed by the transformations.

Our method is based on these two requirements, where a separate map is associated with each domain. For each treatment, the average Wasserstein distance among all pairs of transformed distributions across domains is included in the loss function. Specifically, the average Wasserstein distance is formulated as (3) 2N(N−1)∑i,j=1,i<jNW(Adi(νi),Adj(νj)),

where the coefficient is the normalizing constant, and Adi is the map associated with domain di. Notice that in this article we considered Adi to be an affine transformation, but this restriction is not needed in general. When multiple treatments are considered, the same number of average Wasserstein distances corresponding to the treatments are included in the loss function. Thus, (i) is achieved by minimizing a loss function containing pairwise Wasserstein distances. Compared to the ResNet used in Shaham et al. (2017), we achieve (ii) by early stopping to avoid distorting the distributions and hence destroying the biological signal. Adding a regularization term that penalizes the difference between the transformation and the identity map is also feasible. In “Conclusions”, we will present another possible formulation that aligns more closely with the idea of the Wasserstein barycenter. The general idea of our method is illustrated in Fig. 2 by a concrete example.

Figure 2 Illustration of our approach.

In this case there are three domains (bottom of the plot), each having embeddings for three different treatments. The set of embeddings for treatment i in domain j are represented by Ci,j. Each domain has a map Aj transforming all embeddings in domain j to the combined domain space. Treatments are denoted in shades of magenta while domains are in green. Transformed embeddings are collected into groups each having the same treatment. Our goal is to match the distributions within each such group. To achieve this, we wish to minimize the sum of pairwise Wasserstein distances among those distributions that should be matched. A group that contains embeddings from only one domain is not included in the sum. The top box shows the pairwise Wasserstein losses, whose sum is inserted into the loss function. We use an additional neural network to estimate each Wasserstein distance (not shown) since it is not analytically computable. Moreover, there is either a regularization term or early stopping to preserve the geometry of the original embeddings. In this figure, it is illustrated in its most general form as a function of the transformations Aj. However, we have found that in our setting it was sufficient to use early stopping, which had essentially the same effect.

The Wasserstein distance does not have a closed form except for a few special cases, and must be approximated in some way. The Wasserstein distance is closely related to the maximum mean discrepancy (MMD) approximated in Shaham et al. (2017) using an empirical estimator based on the kernel method. This method requires selecting a kernel and relevant parameters. In our application, we do not have a fixed “target” distribution, so the kernel parameters would have to be updated during training. Instead, we choose to use a method based on the ideas in Arjovsky, Chintala & Bottou (2017) and Gulrajani et al. (2017) to train a neural network to estimate the Wasserstein distance. A similar approach has been proposed in Shen et al. (2017) for domain adaptation. To achieve this, we first apply the Kantorovich–Rubinstein duality to Eq. (1): (4) W(νr,νg)=sup∥f∥L≤1⁡Ex∼νr[f(x)]−Ex∼νg[f(x)]

Here, νr and νg are two probability distributions. The function f is in the space of Lipschitz functions with Lipschitz constant 1. We will call f the “Wasserstein function” throughout this manuscript.

Network architecture

Collectively denote the parameters for the transformations Ad by ΘT. If a particular treatment t is replicated across two or more domains d1, d2,…,dk, the Wasserstein distances among the transformed distributions are estimated for all same-treatment domain pairs. Note that the parameters for estimating the Wasserstein distance for each t and pair di, dj are separate. Collectively denote all the Wasserstein estimation parameters by ΘW. We consider the following loss function (5) L(ΘT,ΘW)=1|T|∑t∈T2|Dt|(|Dt|−1)∑di,dj∈Dt,i<jW(Adi(νi),Adj(νj))+R(ΘT)

where W(Adi(νi),Adj(νj))=sup‖f‖L≤1Ex∼Adi(νi)[f(x)]−Ex∼Adj(νj)[f(x)]

and the term inside the sup operator is called the critic loss. The function R (ΘT) is a regularization term for the learned transformations whose purpose is to preserve the geometry of the original embeddings. For example, R (ΘT) can be proportional to the distance of the transformations from the identity map. Moreover, Dt denotes the domains in which treatment t appears, T denotes all the treatments, and |·| represents the cardinality of a set. In this article, we ignore R entirely and rely on early stopping, as numerical experiments using regularization have given comparable results. The implementation of early stopping is illustrated in “Early Stopping by Leave-One-Compound-Out Cross Validation” by leave-one-compound-out cross validation for our particular dataset to preserve the biological information in the data.

Each Wasserstein function f should be Lipschitz with Lipschitz constant 1. For differentiable functions, this is equivalent to the norm of its gradients being bounded by 1 everywhere. We use an approach based on Gulrajani et al. (2017) to impose a soft constraint on the gradient norm. More specifically, the hard constraint is replaced by a penalty, which is a function of the gradient of the Wasserstein function evaluated at some set of points. The penalty term is weighted by an additional parameter γ. Thus, W(Adi(νi),Adj(νj)) can be approximated by (6) maxΘW⁡[Wt,di,dj(ΘT,ΘW)−gt,di,dj(ΘT,ΘW)],

where, Wt,di,dj is the sample estimate of the critic loss, that is, (7) 1|It,di|∑p∈It,dift,di,dj(Adi(xt,di,p;ΘT);ΘW)−1|It,dj|∑q∈It,djft,di,dj(Adj(xt,dj,q;ΘT);ΘW)

and gt,di,dj is the sample estimate of the gradient penalty, that is, (8) 1|Jt,di,dj|∑z∈Jt,di,djHt,di,dj(z;ΘW)

where, (9) Ht,di,dj(z;ΘW)=(γ(Gt,di,dj(z;ΘW)−1)2if Gt,di,dj(z;ΘW)>10otherwise

(10) Gt,di,dj(z;ΘW)=∥∇ΘWft,di,dj(z;ΘW)∥2.

Each Wasserstein function ft,di,dj in Eq. (7) depends on the parameters ΘW, while each transformation Ad depends on the parameters ΘT. For simplicity, we assume that |It,di|=|It,dj|. This is a reasonable assumption because in practice, the sets It,d are chosen as minibatches in stochastic gradient descent. Since it is impossible to check the gradient everywhere, we use the same strategy as Gulrajani et al. (2017): choose the intermediate points εAdi(xt,di,pk;ΘT)+(1−ε)Adj(xt,dj,qk;ΘT) randomly, where ε ∈ U [0, 1] and pk and qk denote the k-th elements of It,di and It,dj, respectively. The set of intermediate points are denoted by Jt,di,dj. Intuitively, the reason for sampling along these paths is that the Wasserstein function f whose gradient must be constrained has the interpretation of characterizing the optimal transport between the two probability distributions, and therefore it is most important for the gradient constraint to hold in the intermediate region between the distributions. This is motivated more formally by Proposition 1 in Gulrajani et al. (2017), which shows that an optimal transport plan occurs along straight lines with gradient norm 1 connecting coupled points between the probability distributions. Unlike Gulrajani et al. (2017), we impose the gradient penalty only if the gradient norm is greater than 1. Doing so works better in practice for our application. We find that the value of γ = 10 used in Gulrajani et al. (2017) works well in our application, and fix it throughout. This is an appropriate choice since it is large enough so that the approximation error in the Wasserstein function is small, while not causing numerical difficulties in the optimization routine.

Thus, our objective is to find (11) Θ^T,Θ^W=argminΘTargmaxΘWL(ΘT,ΘW).

We use the approach of Ganin & Lempitsky (2015) to transform our minimax problem to a minimization problem by adding a “gradient reversal” between the transformed embeddings and the approximated Wasserstein distances. The gradient reversal is the identity in the forward direction, but negates the gradients used for backpropagation.

Dataset and preprocessing steps

Our analyses are conducted on the image dataset BBBC021 (Caie et al., 2010) available from the Broad Bioimage Benchmark Collection (Ljosa, Sokolnicki & Carpenter, 2012). This dataset corresponds to cells prepared on 55 plates across 10 separate batches, and imaged in three color channels (i.e., stains); for a population of negative control cells, a compound (DMSO) with no anticipated drug effect was applied, while various other drug compounds were applied to the remaining cells. We use the same subset of treatments (a drug compound with a particular dose) evaluated in Ljosa et al. (2013) and Ando, McLean & Berndl (2017). This subset has 103 treatments from 38 drug compounds, each belonging to one of 12 known MOA groups. Both of our datasets rely on methods that isolate cropped images of individual cells, and generate the embeddings on a per-cell basis. A diagram illustration of the dataset is shown in Fig. 3. It is worth pointing out that only the negative control (i.e., DMSO) has replicates across batches. For this reason, the only pairs of distributions we used for finding the domain transformations were controls.

Figure 3 Illustration of the BBBC021 dataset considered in the article.

This dataset corresponds to cells prepared across 10 separate batches, to which DMSO (negative control drug compound, labeled by 0) and other drug compounds (labeled by numbers starting from 1) were applied. We use the same subset of treatments (a drug compound with a particular dose) evaluated in Ljosa et al. (2013) and Ando, McLean & Berndl (2017). This subset has 103 treatments from 38 drug compounds, each belonging to one of 12 known mechanism of action (MOA) groups (shown in the legend with different colors). Each batch (represented by a rectangle) can contain multiple drug compounds with various doses, each represented by a circle with its label colored according to its MOA. If a drug compound has multiple doses in the same batch, the same number of circles with the exact same pattern are plotted.

The treatment is a free label coming from the experiment available for matching treatment replicates across batches (i.e., domains). Multiple drug compounds can share the same MOA, so it is not straightforward to infer the MOA from a drug compound. Sample cell images from the 12 MOA groups are shown in Fig. 1.

The main hyperparameter we considered is the stopping time step for training. This acts as regularization for our model, and prevents biological signal from being lost by collapsing biologically distinct distributions together. We determine the stopping time using one of our metrics, determined over an evaluation set. To mimic the real-world situation of an unknown new compound, the training and evaluation sets are determined using a leave-one-compound-out setting, discussed further in “Early Stopping by Leave-One-Compound-Out Cross Validation.”

We apply our method to embeddings that are generated from this dataset in two different ways: hand-engineered features and features extracted by a pre-trained Deep Neural Network. Since only DMSO was replicated across wells in this dataset, we were able to train only using DMSO.

Embeddings based on hand-engineered features

The embeddings are hand-engineered features of length 453 based on specific features for each cell, generated in Ljosa, Sokolnicki & Carpenter (2012). In the original article, the authors explored a number of ways to preprocess the embeddings. We take the approach of factor analysis that gives the best performance. For each coordinate, the 1st percentile of DMSO-treated cells is set to zero and the 99th percentile is set to 1 for each plate separately. The same transformations are then applied to all drug compounds on the same plate. After this, factor analysis is applied to the transformed embeddings to reduce the dimensionality to 50, to which our nuisance variation correction method is applied. We will refer to the embeddings generated in this way as Preprocessed, and remark this procedure is different from that used for the DNN embeddings presented in “Embeddings Based on Deep Neural Network.”

Embeddings based on deep neural network

We began with the embeddings generated by the pipeline of Ando, McLean & Berndl (2017), which included computing a flatline image for each plate and channel using the 10th percentile.

We compute the embeddings for each cell image using the method in Ando, McLean & Berndl (2017), summarized as follows. For a 128 by 128 pixel crop around each cell for each of the three color channels, a pre-trained Deep Metric Network generates a 64-dimensional embedding vector. The three vectors corresponding to the three color channels are concatenated, forming a 192-dimensional embedding for each cell image. In the principal component analysis (PCA) basis of only negative control embeddings, an affine transformation is found so that the negative control embeddings have mean zero and unit variance (PCA whitening). The same transformation is then applied to the embeddings of all other cells. Note that the TVN method in Ando, McLean & Berndl (2017) includes an additional transformation named CORAL (i.e., correlation alignment), which will be presented and compared to our preprocessing step in “Baselines.” Our nuisance variation correction method is conducted for the embeddings after PCA whitening, which we consider as the preprocessing step for the embeddings from the deep neural network.

Evaluation metrics

Our method is evaluated by three metrics, the first two of which measure how much biological signal is preserved in the transformed embeddings, and the last one of which measures how much nuisance variation has been removed.

k-Nearest neighbor mechanism of action assignment

Each drug compound in the BBBC021 dataset has a known MOA. A desirable property of embedding vectors is that drug compounds with the same MOA should group closely in the embedding space. This property can be assessed in the following way using the ground truth MOA label for each treatment.

First, compute the mean mX of the embeddings for each treatment X in each domain (the negative control is excluded). Find the nearest k neighbors nX, 1, nX, 2,…,nX, k of mX either (i) not belonging to the same compound (abbreviated as NSC) or (ii) not belonging to the same compound or batch (i.e., domain) (abbreviated as NSC NSB), and compute the proportion of them having the same MOA as mX. Our metric is defined as the average of this quantity across all treatment instances X in all domains. If nuisance variation is corrected by transforming the embeddings, we may expect this metric to increase. The reason for excluding same-domain nearest neighbors is to prevent the metric from being interfered by the in-domain correlations.

The nearest k neighbors are found based on the cosine distance, which has already been used in existing literature, and in our numerical experiments, we find that it performs better than the Euclidean distance. Moreover, our k-NN metrics are generalizations of the 1-NN metrics used in Ljosa et al. (2013) and Ando, McLean & Berndl (2017).

Silhouette score on mechanism of action

Cluster validation measures provide another way of characterizing how well drug compounds with the same MOA group together in the embedding space. In our application, each “cluster” is a chosen MOA containing a group of treatments (the negative control is excluded), and each point in a cluster is the mean of embeddings for a particular treatment (i.e., compound and concentration) and domain.

The Silhouette score is one such measure that compares each point’s distance from points in its own cluster to its distance from points in other clusters. Compared to the k-NN metrics as a local metric, Silhouette score is a global metric. It is defined as (12) s(i)=b(i)−a(i)max{a(i),b(i)}

where a(i) is the average distance from point i to all other points in its cluster, and b(i) is the minimum of all average distances from i to all other clusters (i.e., the distance to the closest neighboring cluster) (Rousseeuw, 1987). The Silhouette score ranges between −1 and 1, with higher values indicating better clustering results. As we did with the k-NN metrics, we also use the cosine distance for the Silhouette score.

Domain classification accuracy per treatment

Another metric measures how well domain-specific nuisance information has been “forgotten” (regardless of the biological signal). To achieve this, for each treatment we train a classifier to predict for each embedding the batch (domain) from the set of possible batches (domains) for that treatment. We evaluate both a linear classifier (i.e., logistic regression) and a random forest with 3-fold cross validation. If nuisance variation is corrected, the batch (domain) classification accuracy should decrease significantly. Because only the negative control (i.e., DMSO) has replicates across experimental batches in our dataset, we train and evaluate these two batch classifiers on this compound only.

Domain classification can also be used for testing the hypothesis that two samples were drawn from the same distribution (Lopez-Paz & Oquab, 2016). This gives an alternative interpretation of our batch classification metric.

Cross validation and bootstrapping

Early stopping by leave-one-compound-out cross validation

For the model with either early stopping or a regularization term, the hyperparameters (i.e., the stopping time step or the regularization weight) can be selected by a cross validation procedure to avoid overfitting (see Godinez et al. (2017) for an example). In particular, we apply this procedure to the case of early stopping. Each time, an individual drug compound is held out, and the stopping time step is determined by maximizing the average k-NN MOA assignment metric for k = 1,…,4 on the remaining drug compounds. We can also determine the stopping time step by maximizing the Silhouette score on the remaining drug compounds.

For the embeddings transformed at the optimal stopping time step, we evaluate the k-NN MOA assignment metrics for the held-out compound only. The procedure is repeated for all the compounds, and the k-NN MOA assignment metrics are aggregated across all the compounds. Intuitively, for each fold of this leave-one-compound-out cross validation procedure, the held-out compound can be regarded as a new compound with unknown MOA, and the hyperparameters are optimized over the compounds with known MOAs.

Estimating standard errors of the metrics by bootstrapping

To assess whether the improvements in the three evaluation metrics are statistically significant, we estimate the standard errors of the metrics using a nonparametric bootstrap method. Each time, the bootstrap samples are generated by sampling with replacement the embeddings in each well, and the metrics are evaluated using the bootstrap samples. We repeat the procedure for 100 times, and obtain the standard errors (i.e., standard deviation) of the bootstrap estimates of the metrics.

For the case when using the leave-one-compound-out cross validation discussed in “Early Stopping by Leave-One-Compound-Out Cross Validation”, the stopping time was selected separately for each set of compounds with the left-out compound removed. This way, we are able to evaluate the performance of our method without leaking information about the stopping time from the left-out compound.

Model training

For simplicity, the embedding transformations are assumed to be affine transformations Ad(x) = Mdx + bd, where Md is a d × d matrix and bd is a d × 1 vector. They are initialized to Md = I, bd = 0, since we wish for the learned transformations to be not too far from the identity map. The intuition is that we can treat nuisance variation as small, random, drug-like perturbations resulting from unobserved covariates. An affine transformation is a first-order approximation to a non-linear perturbation in the sense of the Taylor expansion. It is worth mentioning that we do not expect this assumption to hold in general cases, where nonlinear transformations would be more appropriate.

To approximate each of the Wasserstein functions ft,di,dj in Eq. (7), we use a network consisting of a fully connected layer with Softplus activations followed by a scalar-valued fully connected layer. The Softplus activation is chosen because the Wasserstein distance estimation it produces is less noisy than other kinds of activations and it avoids the issue of all neurons being deactivated (which can occur for example, when using ReLU activations).

The dimension of the first fully connected layer is set to 2. Optimization is done using stochastic gradient descent. For simplicity, the minibatch size for each treatment per iteration step is the same and fixed at 100 throughout. Optimization for both classes of parameters ΘT and ΘW is done using two separate RMSProp optimizers. Prior to training ΘT, we use a “pre-training” period of 100,000 time steps to obtain a good initial approximation for the Wasserstein distances. After this, we alternate between training ΘT for 50 time steps and adjusting ΘW for a single time step.

Baselines

We compare our approach to the Preprocessed embeddings for the hand-engineered embeddings and the embeddings transformed by PCA whitening for the DNN embeddings (which we consider to be a preprocessing step), as well as the embeddings transformed by CORAL (Sun, Feng & Saenko, 2016; Ando, McLean & Berndl, 2017) for both types of embeddings following their respective preprocessing steps. Notice we applied our WDN method on the data preprocessed in the same way.

Specifically, by PCA whitening in this article, we mean that the combined controls were scaled to have zero mean and unity variance along each PCA axis, and for the treatments, the same scaling was applied (see “Embeddings Based on Deep Neural Network”).

CORAL applies a domain-specific affine transformation to the embeddings represented as the rows of a matrix Xd from domain d in the following way. On the negative controls only, the covariance matrix across all domains C as well as the covariance Cd in each domain d are computed. Then, all embedding coordinates in domain d are aligned by matching the covariance matrices. It is done by computing the aligned embeddings Xdaligned=XdRd−1/2R1/2. Here Rd = Cd + ηI and R = C + ηI are regularized covariance matrices. Without loss of generality, we specify C = I. The regularization weight η is set to 1, which is the same as that in Ando, McLean & Berndl (2017).

Results

Figure 4 shows the k-NN MOA assignment accuracy as a function of training time steps for NSC and NSC NSB with k = 1, 2, 3 and 4 using our approach. Knowing when to stop training is a nontrivial problem. We observe that for the DNN embeddings, there is an improvement in all the k-NN metrics to some point, for both filtering based on NSC (Fig. 4B) and NSC NSB (Fig. 4D). When continuing to train, these metrics eventually decrease. Meanwhile, the analogous curves using the hand-engineered embeddings (Figs. 4A and 4C) do not improve (is stable at the beginning and then decreases).

Figure 4 Comparison of the k-NN MOA assignment metrics (%) for NSC (i.e., not-same-compound) and NSC NSB (i.e., not-same-compound-or-batch) with k = 1, 2, 3 and 4 over training time steps for the two types of embeddings using our approach (i.e., WDN).

The hand-engineered embeddings are shown in (A) and (C), and the DNN embeddings are shown in (B) and (D). For the DNN embeddings, there is a significant improvement at some time step over the training. This suggests that the biological signal can be improved if the number of training steps is selected appropriately. For the hand-engineered embeddings, there is little improvement in the metrics.

Figure 5 shows the distribution of the bootstrap k-NN MOA assignment metrics for NSC and NSC NSB with k = 1, 2, 3 and 4. We observe that for the DNN embeddings, WDN is at least comparable to CORAL or even sometimes better, and both are generally better than PCA whitening, in preserving MOA-relevant biological information. For the hand-engineered embeddings, however, all perform roughly the same. We also observe that for the DNN embeddings, WDN using the stopping time step based on the Silhouette score outperforms that based on the average k-NN metric. This can be attributed to the Silhouette score being continuous as a function of its inputs, as well as being a global metric. Tables 1 and 2 summarize the NSC and NSC NSB k-NN metrics for the hand-engineered embeddings, respectively, while Tables 3 and 4 summarize the same metrics for the DNN embeddings. The numbers in parentheses represents the standard deviation of the bootstrap estimates.

Figure 5 Distribution of the bootstrap k-NN MOA assignment metrics (%) for NSC (i.e., not-same-compound) and NSC NSB (i.e., not-same-compound-or-batch) with k = 1, 2, 3 and 4 for different methods.

The hand-engineered embeddings are shown in (A) and (C), and the DNN embeddings are shown in (B) and (D). We compare PCA whitening (or Preprocessed for the hand-engineered embeddings) with CORAL, and WDN using two different stopping time steps—based on the average k-NN metric and the Silhouette score. The distribution is shown as a violin plot with three horizontal lines as the 1st, 2nd and 3rd quartiles. For the DNN embeddings, our approach (i.e., WDN) is at least comparable to CORAL or even sometimes better, and both are generally better than PCA whitening, in preserving MOA-relevant biological information. For the hand-engineered embeddings, all perform roughly the same. Using the Silhouette scores instead of k-NN to find the stopping time slightly improved these metrics, which is due to the stopping times becoming more stable using the Silhouette metric.

Table 1 NSC k-NN MOA assignment metrics (%) for hand-engineered embeddings.

For all methods (Preprocessed, CORAL and WDN), we show the bootstrap results. For WDN, we also show the results using two different stopping time steps—based on the average k-NN metric and the Silhouette score. The bootstrap results are represented by the mean and standard deviation (shown in parentheses) across the bootstrap estimates.

Method	Type	1-NN	2-NN	3-NN	4-NN	
Preprocessed	Bootstrap	91.3 (0.8)	86.2 (0.6)	80.7 (0.6)	77.0 (0.6)	
CORAL	Bootstrap	90.4 (0.9)	86.2 (0.7)	80.2 (0.7)	77.4 (0.6)	
WDN	Max Silhouette	92.2	86.4	80.9	76.6	
	Max average k-NN	92.2	86.4	80.9	77.4	
	Bootstrap, max k-NN	91.1 (0.8)	86.0 (0.6)	80.4 (0.8)	76.8 (0.7)	
	Bootstrap, max silhouette	91.1 (1.0)	86.0 (0.6)	79.8 (0.7)	76.4 (0.6)	

Table 2 NSC NSB k-NN MOA assignment metrics (%) for hand-engineered embeddings.

For all methods (Preprocessed, CORAL and WDN), we show the bootstrap results. For WDN, we also show the results using two different stopping time steps—based on the average k-NN metric and the Silhouette score. The bootstrap results are represented by the mean and standard deviation (shown in parentheses) across the bootstrap estimates.

Method	Type	1-NN	2-NN	3-NN	4-NN	
Preprocessed	Bootstrap	73.6 (1.6)	72.9 (1.3)	69.4 (1.1)	67.5 (0.9)	
CORAL	Bootstrap	73.7 (1.7)	72.5 (1.2)	68.2 (0.8)	67.8 (0.8)	
WDN	Max Silhouette	73.9	72.8	69.8	67.3	
	Max average k-NN	73.9	73.3	69.4	67.9	
	Bootstrap, max k-NN	73.6 (1.7)	72.7 (1.3)	69.0 (1.2)	67.3 (1.0)	
	Bootstrap, max Silhouette	73.5 (1.6)	72.3 (1.2)	68.2 (0.9)	67.0 (0.8)	

Table 3 NSC k-NN MOA assignment metrics (%) for DNN embeddings.

For all methods (PCA whitening, CORAL and WDN), we show the bootstrap results. For WDN, we also show the results using two different stopping time steps—based on the average k-NN metric and the Silhouette score. The bootstrap results are represented by the mean and standard deviation (shown in parentheses) across the bootstrap estimates.

Method	Type	1-NN	2-NN	3-NN	4-NN	
PCA whitening	Bootstrap	94.7 (1.0)	92.1 (0.6)	89.1 (0.6)	87.4 (0.5)	
CORAL	Bootstrap	96.8 (0.5)	94.5 (0.5)	91.1 (0.6)	89.0 (0.5)	
WDN	Max Silhouette	97.1	95.1	92.6	90.2	
	Max average k-NN	97.1	95.1	92.6	90.2	
	Bootstrap, max k-NN	96.4 (0.8)	94.2 (0.8)	91.5 (0.7)	89.7 (0.6)	
	Bootstrap, max Silhouette	96.5 (0.7)	94.3 (0.7)	91.8 (0.7)	89.8 (0.6)	

Table 4 NSC NSB k-NN MOA assignment metrics (%) for DNN embeddings.

For all methods (PCA whitening, CORAL and WDN), we show the bootstrap results. For WDN, we also show the results using two different stopping time steps—based on the average k-NN metric and the Silhouette score. The bootstrap results are represented by the mean and standard deviation (shown in parentheses) across the bootstrap estimates.

Method	Type	1-NN	2-NN	3-NN	4-NN	
PCA whitening	Bootstrap	90.2 (1.1)	88.2 (0.9)	84.0 (0.6)	83.4 (0.6)	
CORAL	Bootstrap	92.5 (1.2)	89.4 (0.8)	84.9 (0.6)	84.5 (0.5)	
WDN	Max Silhouette	93.5	91.7	85.8	84.8	
	Max average k-NN	93.5	91.7	85.8	84.8	
	Bootstrap, max k-NN	92.2 (1.2)	90.3 (1.0)	85.2 (0.7)	84.5 (0.6)	
	Bootstrap, max Silhouette	92.6 (0.9)	90.8 (0.8)	85.8 (0.7)	84.8 (0.5)	

The biological signal contained in the embeddings can be also captured by the metric the Silhouette score described in “Silhouette Score on Mechanism of Action.” Figure 6 shows the bootstrap Silhouette scores as a function of training time steps for the hand-engineered and DNN embeddings. For the DNN embeddings, the Silhouette score of WDN significantly increases at the beginning (even above CORAL) and eventually decreases. This suggests that WDN better improves the biological signal than the other methods when the stopping time step is appropriately selected, and over-correction (i.e., being trained for too many time steps) can instead destroy the biological signal. For the hand-engineered embeddings, the Silhouette score only slightly increases at the beginning, reflecting a minor batch effect correction. Tables 5 and 6 compare the bootstrap Silhouette scores at the time step where it is maximized for the hand-engineered and deep-learned features, respectively. For the hand-engineered embeddings, the Silhouette scores are comparable among the methods, while for the DNN embeddings, the Silhouette score of WDN is significantly better than those of the other methods.

Figure 6 Bootstrap Silhouette scores on MOA over training time steps for different methods.

Results for the hand-engineered embeddings are shown in (A) and the DNN embeddings in (B). We compare the methods PCA whitening (or Preprocessed for the hand-engineered embeddings), CORAL and WDN. Silhouette scores for non-WDN methods are independent of the training time steps. The solid line is the mean of the bootstrap Silhouette scores, and the lower and upper bounds (dashed lines for PCA whitening/Preprocessed and CORAL, and error bars for WDN) are mean minus or plus one standard deviation, respectively. The hand-engineered embeddings are shown in (A), and the DNN embeddings are shown in (B). For the DNN embeddings, the Silhouette score of WDN significantly increases at the beginning (even above CORAL) and eventually decreases, suggesting MOA-relevant information can be maximized with selection of the model at a particular time step.

Table 5 Silhouette scores and batch classification accuracy for hand-engineered embeddings.

We compare the methods Preprocessed, CORAL and WDN. The Silhouette score is evaluated over the entire dataset (excluding the negative control), and the batch classification accuracy is evaluated over the negative control, both at the time step where the Silhouette score is maximized. The bootstrap results are represented by the mean and standard deviation (denoted by σ(·)) across the bootstrap estimates.

	WDN	σ (WDN)	CORAL	σ (CORAL)	Preproc.	σ (Preproc.)	
Silhouette score	0.438		0.440		0.436		
Bootstrap Silhouette score	0.429	0.003	0.433	0.003	0.429	0.003	
Bootstrap logistic regression	30.0%	0.8%	33.3%	0.9%	33.9%	0.9%	
Bootstrap random forest	27.3%	0.2%	27.2%	0.2%	29.5%	0.8%	

Table 6 Silhouette scores and batch classification accuracy for DNN embeddings.

We compare the methods PCA whitening, CORAL and WDN. The Silhouette score is evaluated over the entire dataset (excluding the negative control), and the batch classification accuracy is evaluated over the negative control, both at the time step where the Silhouette score is maximized. The bootstrap results are represented by the mean and standard deviation (denoted by σ(·)) across the bootstrap estimates.

	WDN	σ (WDN)	CORAL	σ (CORAL)	PCA	σ (PCA)	
Silhouette score	0.514		0.510		0.504		
Bootstrap Silhouette score	0.505	0.002	0.502	0.002	0.496	0.002	
Bootstrap logistic regression	43.2%	0.6%	66.4%	0.7%	63.6%	1.0%	
Bootstrap random forest	34.6%	1.1%	48.6%	0.6%	45.9%	0.2%	

Other than preserving or enhancing the biological signal, we would like to minimize the ability of using embeddings to distinguish which batch a sample comes from given a treatment. For this reason, we use the metric the batch classification accuracy described in “Silhouette Score on Mechanism of Action.” Figure 7 shows the bootstrap batch classification accuracy using logistic regression and random forest for negative controls as a function of training time steps for the hand-engineered and DNN embeddings. We also include the baseline batch classification accuracy, which is calculated by assigning each embedding coordinate to a Gaussian random variable N(0,1) independently. In this case, there is no batch information whatsoever, so this represents an effective minimum batch classification accuracy level we should expect. For WDN, all the classification accuracy metrics decrease over training time steps, and eventually become closer to the baseline. Tables 5 and 6 compare the bootstrap batch classification accuracy at the time step where the Silhouette score is maximized. We observe that the batch classification accuracy for WDN is significantly lower than those for PCA whitening (or Preprocessed for the hand-engineered embeddings) and CORAL. All of these suggest the effectiveness of our method in removing the batch effect.

Figure 7 Bootstrap batch (i.e., domain) classification accuracy using logistic regression and random forest for negative controls over training time steps for different methods.

Logistic regression results are shown in (A) and (B) and random forests in (C) and (D). The hand-engineered embeddings are shown in (A) and (C), and the DNN embeddings are shown in (B) and (D). We compare PCA whitening (or Preprocessed for the hand-engineered embeddings), CORAL and WDN. Batch classification accuracy for non-WDN methods are independent of the training time steps. The solid line is the mean of the bootstrap batch classification accuracy, and the lower and upper bounds (dashed lines for PCA whitening/Preprocessed and CORAL, and error bars for WDN) are mean minus or plus one standard deviation, respectively. We also include the baseline batch classification accuracy, which is calculated by assigning each embedding coordinate to a Gaussian random variable N(0,1) independently. In this case, there is no batch information whatsoever, so this represents an effective minimum batch classification accuracy level we should expect. The hand-engineered embeddings are shown in (A) and (C), and the DNN embeddings are shown in (B) and (D). All the classification accuracy metrics for WDN decrease over training time steps, and eventually become closer to chance (i.e., baseline), suggesting WDN successfully removes domain-relevant (i.e., batch) information.

Figures 8 and 9 compare the first and last eight principal components of embeddings for negative controls across batches among the three methods PCA whitening (or Preprocessed for the hand-engineered embeddings), CORAL and WDN. Each color corresponds to a batch, and there are ten batches in total. We observe that for the DNN embeddings (Fig. 8, the batch effect is more severe than that for the hand-engineered embeddings (Fig. 9)). This allows us to compare the effect of the different methods side by side in a concrete way. Specifically, we see that for CORAL the greatest batch effect appears in the first few principal components, while for WDN it appears in the last few components. It is possible that for CORAL the nuisance variation between different batches leads to axes of large variance in the combined dataset used to align each individual batch. This would explain why the most batch effect is seen in the first few components. Intuitively, WDN applies an affine transformation that tries forcing batches to align. This would result in moving the batches closer and “shrinkage” along the axes of greatest variation, which is reflected in the batch effect being forced to the smallest principal components.

Figure 8 Principal component comparison for DNN embeddings.

We show the first and last eight PC components for PCA whitening, CORAL and WDN (A–VV). We see for CORAL there is greater batch variation in the first few components, and for WDN in the last few. For CORAL, it may be that the nuisance variation among different batches results in increased covariance along particular axes in the combined dataset, and the embeddings of individual batches become aligned along these directions. For WDN, this represents “shrinkage” in the directions of greatest batch variation, transferring the nuisance variation to the smaller principal components.

Figure 9 Principal component comparison for hand engineered embeddings (A–VV).

These embeddings did not show as much nuisance variation as the DNN embeddings. However, we see similar behavior for CORAL and WDN (see Fig. 8).

While viewing the principal components can be revealing in terms of the concrete behavior of each of the transformations we considered, it may not give a complete picture in terms of how much batch effect is removed. This is reflected in Fig. 10, which shows the proportion of variance explained by each PC component for each method given the two types of embeddings we used. This profile is fairly flat, meaning that there may be meaningful information in both higher and lower principal components. For this reason, to study how much batch effect is removed we also rely on our batch removal metrics as measured by batch classifiers (see Fig. 7).

Figure 10 Ratio of variance explained by each PC component.

The DNN embeddings are shown in (A) and the hand-engineered embeddings are shown in (B). For both DNN embeddings and hand-engineered embeddings, the curve of the portion of variance explained by the principal components is fairly flat. For this reason, in Figs. 8 and 9 we show the distributions resulting from each method for a number of principal components from both head and tail.

Discussion

The framework of our approach is general and extensible. Our method can also utilize information from replicates of multiple treatments across different domains. However, the BBBC021 dataset used does not have treatment replicates across batches, so we have to rely on aligning based on the negative controls only. This means we implicitly assume that the transformations learned from the negative controls can be applied to all the other treatments. We expect our method to be more powerful in the context of experiments where many replicates are present, so that they can all be aligned simultaneously. We expect the transformations learned for such experiments to have better generalizability since it would use available knowledge from a greater portion of the embedding space. Still, we have found for the DNN embeddings that WDN is able to remove the ability to distinguish the batch of origin while preserving or improving biological signal.

Although methods using neural networks tend to be more flexible than traditional methods, they tend to be more difficult to train, in part due to hyperparameter tuning. In the case of our method, we design a minimax problem that once optimized will remove the nuisance variation. However, we must use either early stopping or some form of regularization to prevent collapsing the embeddings together. Although Shaham (2018) does not have the exact same problem in the variational autoencoder setting, they instead need to either use a regularization parameter, or another hyperparameter to balance parts of the loss function associated with the removal of the batch effect and the preservation of the biological signal. Meanwhile, Amodio et al. (2018) defined an explicit map in the latent space, which fixes the percentiles of the distributions to match. While this resolves the hyperparameter ambiguity, it also reduces the flexibility of the method and has to depend on certain assumptions of the latent space for the method to work.

We also remark that additional comparison of recent results is available in Table 2 of Caicedo et al. (2018). Our metrics are somewhat different, so the results are not directly comparable to ours, but one of the higher-performing methods listed there is the result from Ando, McLean & Berndl (2017) to which we have provided a direct comparison in this manuscript.

Conclusions

We have shown how a neural network can be used to transform embedding vectors to “forget” specifically chosen domain information as indicated by our proposed domain classification accuracy metric. The transformed embeddings still preserve the underlying geometry of the space and maintain or even improve the k-NN MOA assignment metrics and the Silhouette score. Our approach uses the Wasserstein distance and can in principle handle fairly general distributions of embeddings (as long as the neural network used to approximate the Wasserstein function is general enough).

We discuss potential future directions below, as well as other limiting issues. One possible extension is to modify the form of the loss function by the following, which would more closely resemble finding the Wasserstein barycenter: (13) ∑i,j=1NW(νi,Adj(νj))

The difference between this and our presented method is that instead of comparing the pairwise transformed distributions, we compare the transformed distributions to the original distributions. One distinct advantage of this approach is that it avoids the “shrinking to a point” problem, and therefore does not require early stopping or a regularization term to converge to a meaningful solution. However, we have not found better performance for the new form of the loss function (Eq. (13)) for the BBBC021 dataset.

The Wasserstein functions were approximated by quite simple nonlinear functions, and it is possible that better results would be obtained using more sophisticated functions to capture the Wasserstein distance and its gradients more accurately. Similarly, The transformations Ad could be generalized from affine to a more general class of nonlinear functions. As in Shaham et al. (2017), we expect ResNet would be natural candidates for these transformations.

We may fine-tune the Deep Metric Network used to generate the embeddings instead of training a separate network on top of its outputs (i.e., embeddings). Another issue is how to weigh the various Wasserstein distances against each other. This might improve the results if there are many more points from some distributions than others (which happens quite often in real applications). Another extension may involve applying our method hierarchically to the various domains of the experiment. For example, we could apply our method on the plate level instead of the batch level only.

Since the k-NN MOA assignment metrics and the Silhouette score are based on the cosine distance, it is possible that better results could be obtained by modifying the metric used to compute the Wasserstein distance accordingly, for example, finding an optimal transportation plan only in non-radial directions.

It is also possible to try applying WDN without the MOA given, which we only use to estimate the stopping time. To this end, one strategy might be estimating the clustering of various treatments instead of relying on given MOA data. However, for this strategy to work, we would also need a way to estimate the number of clusters K. This could cause additional potential issues: If K is estimated before enough batch effect is removed, we may find more clusters than expected when two related treatments appear artificially far because of the batch effect. On the other hand, if we try to estimate K at a late training step, it is possible that too much of the biological signal would be removed resulting in the appearance of too few clusters.

Appendix: Wasserstein vs. Mean-covariance Matching

In a toy example, we look at how minimizing a regularized pairwise 1-Wasserstein distance compares with minimizing a mean-covariance metric. The purpose of this exercise is to demonstrate situations in which higher moments are important for the purposes of matching distributions, and to illustrate benefits of using metrics like the 1-Wasserstein distance.

When matching the mean and covariance, there is generally a free parameter between how important the mean vs. the covariance are. However, in the analysis below we will be concerned with an example in which both can be well-matched, although we will see that there is another sense in which the distributions could be matched better. For this reason we do not weight the mean or variance in the loss function of this method.

In this toy example, we confine the matching to use an affine transformation for each distribution. Each of the two distributions is a Gaussian mixture, with two Gaussians of variance 1 (each will be referred to as its sub-populations). The two Gaussians have a different number of points, N1 = 4,000 and N2 = 6,000. We initialize the two distributions so that the Gaussian with N1 points of the first distribution is closer to the Gaussian with N2 points of the second distribution, and vise versa. The two distributions also have a different offset between their sub-populations (6 and 8). We show our chosen distributions in Fig. 11A.

Figure 11 Toy example comparing matching strategies.

The initial two distributions to match are shown in (A), and the results from the two matching strategies are shown in (B) for mean-covariance matching and (C) and Wasserstein distance minimization.

Optimizing how much the two distributions match involves multiple tradeoffs. (1) We do not want the transformations to be too strong, collapsing both distributions. For this reason, we introduce a regularization term. In our main work, we use early stopping, which plays a similar role. (2) The two distributions should align as well as possible. Towards this goal, there are multiple intuitive tradeoffs as well. We would like each sub-population to match the location of the corresponding sub-population in the other distribution, but also we would like to avoid a distortion between the shapes of the resulting sub-populations. In this example these are two competing goals, since shrinking the distribution with a wider gap between the two sub-populations will necessarily distort their shape.

The two optimization strategies we consider produce visually different outcomes. Since the optimal regularization for the two strategies may be on a different scale, we adjusted the regularization for each strategy and examined the results.

When the regularization is made small (≤0.1), the Wasserstein experiment flips the two sub-populations for one of the distributions. This is reasonable since the sub-populations initially close to each other in the two distributions have imbalanced sizes. However, we will restrict our analysis to the regime when this flip does not happen. Meanwhile, with even very small regularization (0.00005) the mean-covariance matching experiment does not flip the two populations, and the results look much the same as with strong regularization (0.5). In the case of very small regularization, we find this experiment matches the means and variances very closely (O(1e−5), without altering the overall scale). Interestingly, the mean-covariance matching experiment does eventually flip the distributions at regularization somewhere between 0.000004778 and 0.000004779.

When regularization is strong (>0.3), neither distributions flip horizontally. In both experiments, the larger sub-populations move towards the center compared to the smaller sub-populations. However, in the Wasserstein experiment there is visually greater overlap between the resulting distributions. The reason in this difference is that the Wasserstein method is able to account for high-order moments, especially including the skew introduced because of the difference in the sub-population size. When trying this experiment with equally-sized sub-populations, the results of the two experiments are much more similar.

To compare the matching quality, we consider how well each sub-population has been matched individually for the two methods. To account for the overall scale changing, we use the metric.

(14) Δ1,2=∥μ1−μ2∥σ12+σ22

where μi and σi are the mean and standard deviation of sub-population i.

The mean-covariance matching produces scores of 1.0 and 0.91, and the Wasserstein minimization 0.41, 0.36 for the two corresponding sub-populations, agreeing with the visual results.

We would like to thank Mike Ando, Marc Coram, Marc Berndl, Subhashini Venugopalan, Arunachalam Narayanaswamy, Yaroslav Ganin, Luke Metz, Eric Christiansen, Philip Nelson, and Patrick Riley for useful discussions and suggestions.

Additional Information and Declarations

Competing Interests

Author Contributions

Data Availability

The authors are employes of Google LLC.

Gil Tabak conceived and designed the experiments, performed the experiments, analyzed the data, prepared figures and/or tables, authored or reviewed drafts of the paper, and approved the final draft.

Minjie Fan conceived and designed the experiments, performed the experiments, analyzed the data, prepared figures and/or tables, authored or reviewed drafts of the paper, and approved the final draft.

Samuel Yang conceived and designed the experiments, prepared figures and/or tables, authored or reviewed drafts of the paper, and approved the final draft.

Stephan Hoyer conceived and designed the experiments, authored or reviewed drafts of the paper, and approved the final draft.

Geoffrey Davis conceived and designed the experiments, authored or reviewed drafts of the paper, and approved the final draft.

The following information was supplied regarding data availability:

Our code has been made open-source and is available at the following repository: Tabak G, Fan M. 2019. Correcting for Batch Effects Using Wasserstein Distance.

https://github.com/google-research/google-research/tree/master/correct_batch_effects_wdn.

We used the data published in Ljosa, Sokolnicki & Carpenter (2012).

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
