# Peer review of "Correcting nuisance variation using Wasserstein distance"

_PeerJ, doi:10.7717/peerj.8594_

## Round 0.1 · original submission · Major Revisions

While the reviewers were generally positive, a number of concerns and points of clarification were raised. In particular, both reviewer 1 and 3 had questions regarding stopping parameters and related criteria. Questions as to the relationship between TVN and CORAL were also raised and would benefit from clarification. Finally, addressing questions involving feature extraction brought up by reviewer 3 would seem to help in providing an improved picture of this approach for the reader.

Reviewer 1 ·

Basic reporting

Well written paper, with clear explanations and easy to follow flow. Minor typos need to be fixed.

Experimental design

* I found somewhat confusing the procedure used to identify the early stopping parameter of the model. In section 2.4, the paper says "The MOA information is used for early stopping in model training", and in section 2.6.1 it reads "the hyperparameters are optimized over the compounds with known MOAs". Since the leave-one-compound-out validation is used for evaluation, how exactly is this evaluation separated from hyperparameter tuning?
* Boosting is employed to estimate variance of the results by sampling with replacement the embeddings in each well. Can you clarify if you mean single-cell embeddings, image-based embeddings or full well embeddings? This is necessary to understand the extent of the variation of the bootstrapping results.
* In this dataset, only DMSO wells appear in multiple batches, but the method can incorporate any treatments that have replicates across batches. Can you confirm that you trained the model on DMSO only for the experiments reported in the paper?
* Can you clarify whether the embedding transformation trained for the experiments in this paper is a neural network or an afine (linear) transformation?
* What is the difference between TVN and CORAL in the context of this paper? According to Ando et al. 2017, the TVN method is based on CORAL corrections as well.
* The WDN method is applied to embeddings after TVN. Why is TVN necessary? Ando et al. 2017 showed that TVN can effectively remove the unwanted variation from embeddings. The WDN method should be also evaluated on embeddings without TVN to assess the applicability as an independent strategy.

Validity of the findings

* It is surprising that CORAL performs better than TVN in the results presented in this paper. Can you comment on this? (also, see comment about difference between TVN and CORAL above).
* The method is trained leaving one compound out, and selecting the early stopping parameter for each compound. I find it confusing to train and evaluate the model at the same time, specially when one critical hyperparameter needs to be selected at each iteration of the evaluation. Can you comment on that?
* It would be nice to present a table with results reported by others to help the reader compare differences with other approaches that use the same dataset.
* Is the optimal stopping step the same for all metrics evaluated? Is it the same for all treatments? Can you give some more information and analysis about this? This is specially important given that the parameter appears to be a critical choice in the method.
* The batch classification plot (Fig 7) seems to indicate that the method keeps removing nuisance variation after the "optimal early stopping" point. Why is it that this improves but MOA assignment decreases?

Reviewer 2 ·

Basic reporting

The authors presented a method of correcting unwanted variation of cellular imaging data while preserving relevant biological information. The Wasserstein distance is used and evaluated the performance compared to other competing methods. Although some results show the advantage of this proposal, I have some concerns about target audience for application, novelty of method (selling points of 1-Wasserstein distance) and supporting examples.

Experimental design

Target audience: I was wondering while reading this article about who will read this paper and further use the proposed method in what situations. The author mentioned it is a general framework but I see this paper is an application to introduce a 'relative new' mathematical idea to a certain situation like cell imaging data in drug development. If you want to sell this method to a drug industry, the motivating example has to be concrete but at the same time popular enough to attract the audience. I suggest the author arguing the example presented here (e.g., Figure 2, page 7) that are very common, more than a case study, and promising datasets in the future as well, so that the readers have a reason to use this method or similar others. Including a second example is a good idea, too. Otherwise, this work is less valuable from an application standpoint. So far, there is a doubt what target audiences this paper can reach.

Validity of the findings

Novelty/Benefit of the method: The authors have to convince the readers why it has to be a Wasserstein distance. Two advantages explained (lines 90 - 92 ) is not fully convincing 1) non-linear removal of batch effects (multi-modal or non-Gaussian data); This advantage should be explained in the text (not only just saying it), theoretically or empirically, through data simulation, toy example, or real data example, which are not sufficiently provided in this work; 2) Adjusting the entire distribution more than two batches - This reason is misconstructed. Some paper like Benito et al (2004) used only two batch problem but its application is not limited to two-batch. Other batch adjustments methods, as referred in Johnson et al (2007), Leek and Storey (2007) and Lee at al (2014) have the approach of multi-batch adjustment, either sequential or simultaneous way. The reason why some methods mentioned `target' batch or `source' of variation (lines 111-112) is rather a biological reason, not a mathematical reason, so it is the strength of their approach when extra information is available from the complex experiments, and importantly these methods could be used in a way of ``equal footing," when there is no biological information available to consider. Reversely, we could question the author if the proposed Wasserstein method can correct the nuisance variation with different weights across different quality of batches if any. Then, it could be a totally different argument, spinning around a weakness of the proposed method. I wish the authors could provide more convincing reasons over other methods.

Distance measures: One way of selling the Wasserstein method is the distance measure itself. You may consider rephrase (lines 143 - 155) or develop stronger argument against other distance metrics, such as 2- Wasserstein distance, Euclidean distance, Mahalanobis distance (or regularized Mahalanobis distance for high dimension), or Cramer distance (as mentioned p.116). Perhaps the cell image data is high-dimensional. Can 1-Wasserstein distance overcome a large dimension situation? Again, the readers need to be convinced of why it has to be a 1-Wasserstein distance.

Philosophy of batch bias removal: The authors suggested three ways of evaluation, two of which measure how much biological signal is preserved in the transformed embeddings, and the other is how much nuisance variation has been removed. There is a philosophical concern for the first two approaches. We could always look at how much the biological signal of interest turned out to be, as a result of background noise removal. However, we should not be using this as a definite standard for the choice of batch removal because this potentially contains a double-dipping problem. The method developed so that preserve the biological difference are meant to display the biological difference, because it was preserved! It is ideal to blindside the biological information and it might result in removing the biological signal as well (and it could be the right thing). For this reason, I would rather recommend the authors to focus on the similarity among batches to evaluate the effectiveness of batch bias removal. The presented evaluation in Figure 8, displaying first two PCs is not enough (How much variation by first two PC are explained?; PC scatter plot in two-dimensional space is better than histogram). Maybe you could consider some statistical testing for the equality of the means, covariance matrices or distributions.

Reviewer 3 ·

Basic reporting

No comment.

Experimental design

No comment.

Validity of the findings

No comment.

Additional comments

The authors propose an approach for normalizing vectors derived from
high-content images monitoring cells subjected to chemical treatments
seeded in plates prepared across different experimental batches. The
approach exploits the fact that vectors from the same treatment
ought to be similar across batches. The approach accordingly minimizes
pairwise distances among the high-dimensional distributions entailed
by the vectors of different batches of the same treatment. The
distance calculation is approximated by a neural network. The authors
test their method on vectors derived
through both conventional and deep learning feature extractors on the
well-known BBBC021 image dataset

The problem of registering vectors onto a common space is an important
one, esp. given the fact that deep learning feature extractors are
susceptible to batch effects.

* One main suggestion is to acknowledge the fact that the idea of
assuming that feature vectors from the same treatment look the same
has been used before, esp. in the context of computing features for
high content images through deep learning approaches, see, e.g.,:
- Caicedo et al. Weakly Supervised Learning of
Single-Cell Feature Embeddings, CVPR, 2018
- Godinez et al. Unsupervised phenotypic analysis of cellular images
with multi-scale convolutional neural networks, bioRXiv, 2018

* Prior to feature extraction with conventional methods, an
illumination step is typically carried out (see, e.g., Singh, S et al. “Pipeline
for illumination correction of images for high-throughput microscopy.”
Journal of microscopy vol. 256,3 (2014)). Since the proposed approach
does not bring much of an improvement for conventional feature
extraction schemes, one wonders whether the reason for this is that
conventional features are applied on illumination-corrected images.
It would be great if the authors could comment on this. One also
wonders whether a) such an illumination correction was carried out in
this study prior to the application of the deep learning feature
extractors, and b) whether, after such a correction, the proposed
approach would still bring an improvement w.r.t. the alignment of the
deep learning-based vectors. It would be great if the authors could
show an improvement even after carrying out an illumination correction
on the images.

* The approach is certainly sensitive to the training time, as the
longer one trains, the higher the likelihood that the learned
transformation would remove subtle but important phenotypic
variations. The early stopping criterion proposed in the paper
requires some type of prior phenotypic knowledge regarding the
dataset, which in a more realistic setting, is hard to come by (often,
the MOA of most compounds in an exploratory scenario is not
known). Are there other early stopping criteria requriring no prior
phenotypic knowledge that could be used for this method?

---

## Round 0.2 · Minor Revisions

Thank you for addressing the reviewer comments. There are just a couple of minor reviewer comments that could help clarify the results if addressed. The link to associated code should also be provided.

Reviewer 1 ·

Basic reporting

The paper presents a theoretical framework for batch correction with one specific instantiation using the Wasserstein distance as the metric to match distributions of data points. The paper reads well.

Experimental design

The experiment is limited to one dataset, but the metrics used for performance evaluation and the experimental protocols are well designed.

Validity of the findings

The results seem inline with previous reports on the same dataset. The authors provide source code for reproducing the results on publicly available features. The methods seems to be better suited for deep learning features computed with pretrained neural networks.

Additional comments

The link to the code could be included in the manuscript.

Reviewer 3 ·

Basic reporting

The usage of the TVN term is still confusing, as it only represents one step of the original TVN method. I suggest using the term 'PCA whitening' instead of TVN.

Line 128: "Our method incorporates all replicates of a given treatment on an equal footing, which can be advantageous over using only the controls." I am now finding this contradictory since only the DMSO controls are used to learn the domain transformations. I suggest removing this sentence.

Lines 247 to 250 are duplicate sentences.

Experimental design

No comment

Validity of the findings

The inclusion of the bootstrap results is a great addition to the paper. The term 'vanilla' should be explained in the main text, though. It's not clear, for example, whether the vanilla result is just another sample in the bootstrap run? Or is the vanilla result using all embeddings? Also, the vanilla and boostrap results are pretty similar. It would aid for legibility to select only one type of result for presentation and discussion, esp. if the conclusions remain the same. The bootstrap ought to be more statistically informative, so I'd lean towards those results.

---

## Round 0.3 · accepted · Accept

Thank you for addressing the reviewer questions and congratulations again.